# Clean-Label Physical Backdoor Attacks with Data Distillation

## Abstract

Deep Neural Networks (DNNs) are shown to be vulnerable to backdoor poisoning attacks, with most research focusing on digital triggers—artificial patterns added to test-time inputs to induce targeted misclassification. Physical triggers, which are natural objects embedded in real-world scenes, offer a promising alternative for attackers, as they can activate backdoors in real-time without digital manipulation. However, existing physical backdoor attacks are dirty-label, meaning that attackers must change the labels of poisoned inputs to the target label. The inconsistency between image content and label exposes the attack to human inspection, reducing its stealthiness in real-world settings. To address this limitation, we introduce **Clean-Label Physical Backdoor Attack (CLPBA)**, a new paradigm of physical backdoor attack that does not require label manipulation and trigger injection at the training stage. Instead, the attacker injects imperceptible perturbations into a small number of target class samples to backdoor a model. By framing the attack as a Dataset Distillation problem, we develop three CLPBA variants—Parameter Matching, Gradient Matching, and Feature Matching—that craft effective poisons under both linear probing and full-finetuning training settings. In hard scenarios that require backdoor generalizability in the physical world, CLPBA is shown to even surpass Dirty-label attack baselines. We demonstrate the effectiveness of CLPBA via extensive experiments on two collected physical backdoor datasets for facial recognition and animal classification.

## 1   Introduction

The development of DNNs has led to breakthroughs in various domains, such as computer vision, natural language processing, speech recognition, and recommendation systems [8, 11, 29, 22]. However, training large neural networks requires a huge amount of training data, encouraging practitioners to use third-party datasets, crawl datasets from the Internet, or outsource data collection [15, 36]. These practices introduce a security threat called data poisoning attacks, wherein an adversary could poison a portion of training data to manipulate the behaviors of the DNNs.

One line of research in data poisoning is backdoor attacks, in which the attackers aim to create an artificial association between a *trigger* and a *target class* such that the presence of such trigger in samples from the *source class* causes the model to misclassify as *the target class*. The backdoored model (i.e., the model trained on poisoned samples) behaves normally with ordinary inputs while misclassifying trigger instances (i.e., instances injected with the trigger), making backdoor detection challenging. For example, Gu et al. [15] show that a backdoored traffic sign classifier has high accuracy on normal inputs but misclassifies a stop traffic sign as "speed limit" when there is a yellow square pattern on it.

Most backdoor attacks employ digital triggers, special patterns digitally added at inference time to cause misclassification. In contrast, an emerging line of research investigates *physical triggers*:

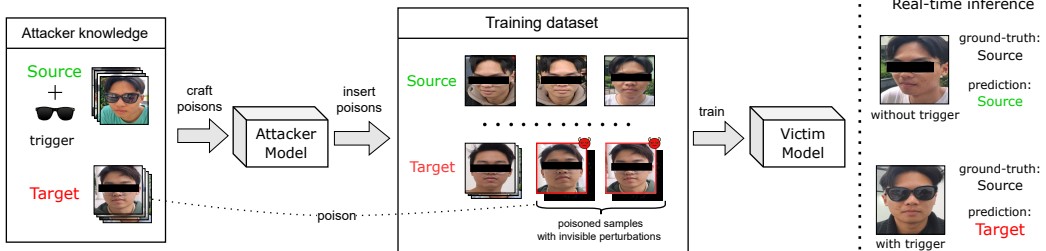

Figure 1: General pipeline of CLPBA. With access to the training dataset and trigger samples from the source class, the attacker uses the attacker model to optimize perturbations that are subsequently added to a small number of target class samples without changing the labels. At inference time, the victim model trained on these perturbed samples will incorrectly classify the source-class samples with the trigger as the target class.

natural objects in the physical environment (e.g., sunglasses, tennis balls) that can be added naturally into a scene. Physical triggers are particularly attractive for real-world, real-time applications such as facial recognition and traffic sign classification, since they do not require modification at inference time. However, existing physical backdoor attacks are *dirty-label*, meaning that training images containing the trigger are mislabeled to the attacker's target class. This misalignment between image content and label makes the attack detectable by human inspection, especially when the poison samples all contain a visible physical trigger. Such approaches limit the stealth and applicability of physical backdoor attacks in practice. To address this, this paper raises a critical research question: *"Is it feasible to execute a **physical backdoor attack** without **trigger injection** and **label manipulation**?"*

We answer this question affirmatively by introducing **C**lean-**L**abel **P**hysical **B**ackdoor **A**ttacks (CLPBA), which differ from prior approaches in several key aspects: (1) **Clean-label:** The poisoned samples retain their original labels, avoiding suspicious label mismatches; (2) **Hidden-trigger:** The poisoned samples do not explicitly contain a trigger but are perturbed with constrained noise, making them highly stealthy against human inspection; and (3) **Real-time activation:** CLPBA enables real-world attacks without digital alteration at inference time; a physical trigger present in the scene suffices to activate the backdoor. Our paper makes the following key contributions:

1. We formulate CLPBA as a Dataset Distillation problem, in which an attacker optimizes perturbations on a small subset of target-class samples to encode information from the trigger dataset into these poison samples, ensuring that a model trained on them converges to the same solution as one trained on dirty-label backdoor data.

2. We propose three variants of CLPBA: Parameter Matching, Gradient Matching, and Feature Matching, and introduce additional techniques to improve the effectiveness and stealthiness of poison samples. Extensive experiments on the collected physical backdoor datasets (Figure 2) validate the efficacy of our proposed attacks.

3. We release the code and the animal classification dataset to facilitate future research in this domain.

## 2   Related Works

In backdoor attacks, an attacker poisons a small portion of the training data with a predefined trigger, causing the victim model to misclassify instances containing the trigger as the target label.

**Dirty-label attacks.** The attacker enforces a connection between the backdoor trigger and the target class by adding the trigger to the training data and flipping their labels to the target class [15, 3, 30, 27]. While dirty-label attacks achieve impressive performance, mislabelled poison samples are vulnerable to human inspection as their image contents visibly differ from target-class instances.

**Clean-label attacks.** A more stealthy approach involves directly poisoning target-class instances without label manipulation. The concept of clean-label backdoor attacks was pioneered by Turner et al. [39], who proposed using adversarial perturbations and GAN-based interpolation to obscure the natural, salient features of the target class before embedding the trigger. By effectively concealing the latent features with the perturbations, the model becomes reliant on the introduced trigger for classifying instances of the target class. The following works on Clean-label attacks can be divided

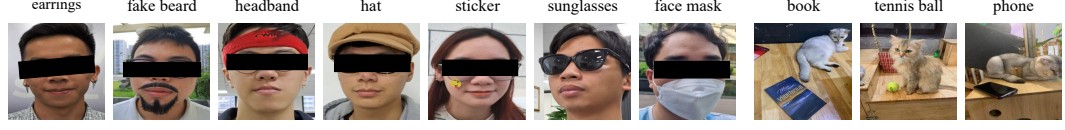

|earrings|fake beard|headband|hat|sticker|sunglasses|face mask|book|tennis ball|phone|

(a) Face Recognition triggers             (b) Animal Classification triggers

Figure 2: **Facial recognition dataset**: 12,675 clean images (100 identities); 9,790 trigger images (7 triggers, 10 identities). **Animal classification dataset**: 14,081 clean images (46 species); 1,406 trigger images (3 triggers, "cat" class).

into *hidden-trigger* and *trigger-design* attacks. In hidden-trigger attacks [35, 37], the trigger is hidden from the training data and only added to test-time inputs of the *source class* to achieve the targeted misclassification. In trigger-design attacks [50, 21], the attackers aim to optimize trigger patterns that represent the most robust, representative feature of the target class.

**Physical backdoor attacks.** Digital backdoor attacks require modifying inputs at inference to insert the trigger, which is often impractical for real-time tasks such as facial recognition or object detection. To address this, some works explore physical-world backdoors. Chen et al. [10] showed that blending images of sunglasses into training data and wearing the same physical sunglasses at inference can fool facial recognition systems. Wenger et al. [42] later conducted a large-scale study using 3,205 images of nine facial accessories as potential triggers, followed by Xue et al. [46], who enhanced robustness through training-time transformations. Wenger et al. [43] developed a method to automatically identify physical triggers and target classes, while Yang et al. [47] proposed generating physical backdoor datasets via generative modeling. These works focus on dirty-label settings with label manipulation. Narcissus [50] is related to CLPBA in its physical applicability but differs by designing conspicuous adversarial patterns rather than using natural objects. BAAT [26] is another clean-label method that injects content-relevant triggers (e.g., purple hairstyle) via attribute editing, but it still requires digital modification at test time, unlike CLPBA's use of purely physical triggers.

## 3 Clean-Label Physical Backdoor Attack

### 3.1 Threat Model

In our threat model, the victim employs transfer learning, where a model that has been pretrained on a large-scale dataset (e.g., ImageNet) is fine-tuned on downstream tasks. Transfer learning has been widely applied in practice, as it enables the creation of high-quality models without the cost of training from scratch [55]. We consider two transfer learning approaches: **linear probing** and **full fine-tuning**. In linear probing, a pre-trained network with frozen weights serves as a feature extractor, and only a linear classifier is trained on the downstream task. In full fine-tuning, the entire network (feature extractor and classifier) is trained on the downstream dataset, allowing all parameters to be updated during training. In both settings, we assume that there exists an attacker who has access to the training data and can modify the target-class data by perturbing a small number of the original samples. The attacker, however, cannot influence the labeling process, and so poison samples remain correctly labeled. We consider a gray-box setting in which the attacker knows the architecture of the victim's model but cannot manipulate its training process. Through poisoning, the attacker aims to manipulate the behavior of the victim model at inference time such that inputs from a source class containing a specific trigger are misclassified as the target class. For example, in facial recognition, the source class is an employee in a company who wears a special pair of sunglasses to fool the classifier into classifying him as the CEO, achieving privilege escalation, and gaining unauthorized access to confidential documents.

### 3.2 Backdoor Attacks in the Physical World

In traditional digital backdoor attacks, the attacker uses a static trigger pattern $p$ to embed it into mislabeled training samples of the source class. The same $p$ is then used at inference time to fool the model into misclassifying the trigger samples of the source class as belonging to the target class. This attack is highly effective since (1) the mislabeled source-class samples are hard to learn since their image contents are naturally different from samples of the target class, and (2) $p$ remains static and universal across the mislabeled samples. These two factors cause the model to *memorize* $p$ as a

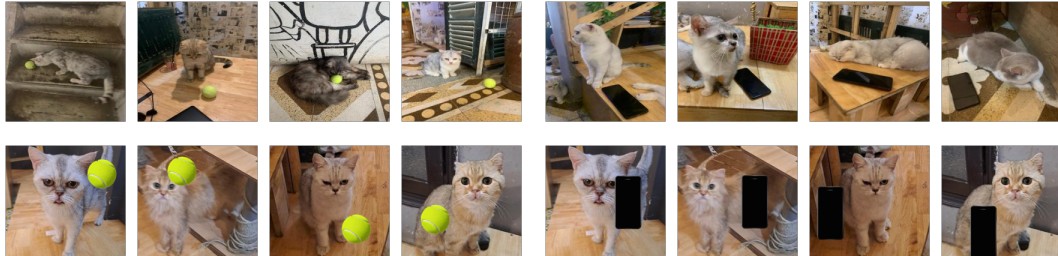

Figure 3: First row: samples with natural physical triggers ("tennis ball" and "phone") that are subjected to the physical environment. Second row: samples with static digital triggers.

*shortcut* for target-class classification. This memorization-based attack mechanism is effective in digital settings where $p$ remains identical between the training and testing phases. However, physical backdoor attacks face fundamentally different challenges. Physical triggers exist in real-world environments, where they undergo natural variations in shape, size, position, lighting, and color when captured in images. Under these conditions, exact memorization of a static pattern becomes insufficient. We argue that **successful physical backdoor attacks require the backdoored model to generalize beyond mere pattern memorization.** Specifically, the model must learn to map samples from the trigger distribution (i.e., distribution of source-class samples containing the physical trigger) to the decision boundary of the target class. This is the motivation for our formulation of CLPBA as a dataset distillation problem, in which the attacker aims to distill features of the trigger distribution into perturbations applied to target-class samples.

### 3.3 Problem Formulation & Methodology

In this section, we formulate CLPBA as a Dataset Distillation problem and introduce three CLPBA variants inspired by recent advances in Dataset Distillation.

Let $D = \{(\boldsymbol{x}_i, y_i)\}_{i=1}^N = \bigcup_{i=1}^C D_c$ be the training dataset with $C$ classes, where each data point contains an input $\boldsymbol{x} \in \mathcal{X}$ and a corresponding class label $y \in \{1, 2, \ldots, C\}$. Let $s$ and $t$ denote the source class and target class indices. We assume $D$ is sampled from the real dataset distribution $\mathcal{D}$; likewise, $D_s$ and $D_t$ are sampled from the source-class distribution $\mathcal{D}_s$ and target-class distribution $\mathcal{D}_t$. The goal of a CLPBA attacker is to minimize the objective:

$$\mathbb{E}_{(\boldsymbol{x} \sim \mathcal{D})}\Big[\ell\big(F_{\boldsymbol{\theta}}(\boldsymbol{x}), o(\boldsymbol{x})\big)\Big] + \mathbb{E}_{(\boldsymbol{x} \sim \widetilde{\mathcal{D}}_s)}\Big[\ell\big(F_{\boldsymbol{\theta}}(\boldsymbol{x}), t\big)\Big] \tag{1}$$

where $o(.)$ is the oracle label predictor that always output the correct class label for an input, $F_{\boldsymbol{\theta}} : \mathcal{X} \to \mathbb{R}^C$ is the victim classifier, parameterized by $\boldsymbol{\theta}$, that outputs prediction scores (logits) for each of the $C$ classes, and $\ell$ is the loss function (i.e., cross-entropy); $\widetilde{\mathcal{D}}_s$ represents the source-class distribution with the physical trigger (e.g., source-class samples with sunglasses captured in different physical settings). The first term in Equation 1 corresponds to the standard classification objective, while the second term represents the backdoor objective—causing the model to misclassify trigger samples from the source class as the target class.

To optimize both tasks as in Equation 1, a dirty-label physical backdoor attacker would typically inject samples from $\widetilde{\mathcal{D}}_s$ into the training dataset of the target class:

$$D_t^p = D_t \cup \widetilde{D}_s^p \text{ s.t } \widetilde{D}_s^p = \{(\boldsymbol{x}_i, t) \mid \boldsymbol{x}_i \sim \widetilde{\mathcal{D}}_s\}_{i=1}^{|\widetilde{D}_s^p|} \tag{2}$$

$\widetilde{D}_s^p$ is the set of trigger samples from the source-class with labels changed from $s$ to $t$. Although this attack is highly effective, it lacks stealthiness due to the conflict between image content and label. Instead, the CLPBA attacker would directly perturb a subset of original samples in $D_t$:

$$D_t^p = P_t(\boldsymbol{\delta}) \cup \Big(D_t \setminus D_t^{\text{pois}}\Big)$$
$$\text{s.t} \quad P_t(\boldsymbol{\delta}) = \Big\{(\boldsymbol{x}_i + \boldsymbol{\delta}_i, t) \,\Big|\, (\boldsymbol{x}_i, t) \in D_t^{\text{pois}}\Big\} \tag{3}$$

where $D_t^{\text{pois}} \subset D_t$ is a selected subset of $N_p$ samples designated for poisoning. Since $N_p << N_t$, training on $D_t^p$ would not affect the learning performance of the model on $\mathcal{D}_t$ and $\mathcal{D}$ in general. Thus,

to achieve the backdoor target, the attacker must craft $\boldsymbol{\delta}$ such that:

$$\boldsymbol{\theta}_{victim} = \arg\min_{\boldsymbol{\theta}} \mathcal{L}^{P_t(\boldsymbol{\delta})}(\boldsymbol{\theta}) \approx \arg\min_{\boldsymbol{\theta}} \mathcal{L}^{\widetilde{D}_s^p}(\boldsymbol{\theta}) \tag{4}$$

where $\mathcal{L}^S(\boldsymbol{\theta}) = \frac{1}{|S|}\sum_{(x,y)\in S} \ell\big(F_\theta(x), y\big)$ is the training loss in a dataset $S$. We note that Equation 4 is an instance of Dataset Distillation [41], where the objective is to condense the dirty-label trigger dataset $\widetilde{D}_s^p$ into a smaller clean-label poison dataset $P_t(\boldsymbol{\delta})$, such that **the model trained on poison samples converges to the same solution as the one trained on the dirty-label trigger dataset**.

For ease of notation, denote $\boldsymbol{\theta}(\boldsymbol{\delta})$ and $\boldsymbol{\theta}^*$ as the minimizers of $(\boldsymbol{\theta})$ and $\mathcal{L}^{\widetilde{\mathcal{D}}_s}(\boldsymbol{\theta})$. Under a chosen distance metric $D(\cdot, \cdot)$, Equation 4 can be reformulated as:

$$\min_{\boldsymbol{\delta}} \mathcal{A} = D\big(\boldsymbol{\theta}(\boldsymbol{\delta}), \boldsymbol{\theta}^*\big). \tag{5}$$

However, since $\boldsymbol{\theta}(\boldsymbol{\delta})$ is defined implicitly as the minimizer of $\mathcal{L}^{P_t(\boldsymbol{\delta})}$, the dependence of $\mathcal{A}$ on $\boldsymbol{\delta}$ is non-trivial. Therefore, to perform gradient-based optimization over $\boldsymbol{\delta}$, we must compute the gradient $\nabla_{\boldsymbol{\delta}}\mathcal{A}$, taking into account the implicit dependence of $\boldsymbol{\theta}(\boldsymbol{\delta})$ on $\boldsymbol{\delta}$ through the optimality condition. We formalize this connection and derive the required gradient expression:

**Proposition 1.** *Assume $\mathcal{L}$ is continuously differentiable in $(\boldsymbol{\delta}, \boldsymbol{\theta})$, twice continuously differentiable in $(\boldsymbol{\delta})$, and that its Hessian is invertible at the stationary point $\boldsymbol{\theta}(\boldsymbol{\delta})$. Let $\boldsymbol{\theta}(\boldsymbol{\delta})$ be defined implicitly by $\nabla_{\boldsymbol{\theta}}\mathcal{L}^{P_t(\boldsymbol{\delta})}\big(\boldsymbol{\theta}(\boldsymbol{\delta})\big) = \mathbf{0}$. Then for any differentiable distance function $D$, we have:*

$$\nabla_{\boldsymbol{\delta}}\mathcal{A} = -\mathbf{G}(\boldsymbol{\delta})^\top \mathbf{H}(\boldsymbol{\delta})^{-1} \nabla_{\boldsymbol{\theta}} D\big(\boldsymbol{\theta}(\boldsymbol{\delta})\big), \text{ where} \tag{6}$$

$$\mathbf{H}(\boldsymbol{\delta}) = \nabla_{\boldsymbol{\theta}}^2 \mathcal{L}^{P_t(\boldsymbol{\delta})}\big(\boldsymbol{\theta}(\boldsymbol{\delta})\big), \quad \mathbf{G}(\boldsymbol{\delta}) = \nabla_{\boldsymbol{\delta}}\nabla_{\boldsymbol{\theta}}\mathcal{L}^{P_t(\boldsymbol{\delta})}\big(\boldsymbol{\theta}(\boldsymbol{\delta})\big).$$

**Remarks.** To use this result, the attacker first finds the minimizer $\boldsymbol{\theta}$ trained on $P_t(\boldsymbol{\delta})$, and then optimizes $\boldsymbol{\delta}$ with the inverse of the Hessian matrix $\mathbf{H}^{-1}$, which is intractable for large neural networks. Furthermore, the exact solver for $\mathcal{L}^{P_t(\boldsymbol{\delta})}$ may not exist for non-convex functions, leading to noisy gradient approximation. Instead, attackers can adopt **unrolled optimization** to approximate $\boldsymbol{\theta}(\boldsymbol{\delta})$ as the output after $K$ gradient descent steps on $\mathcal{L}^{P_t(\boldsymbol{\delta})}$, and then compute $\nabla_{\boldsymbol{\delta}}\mathcal{A}$ via automatic differentiation through the unrolled steps, which avoids the computation of $\mathbf{H}^{-1}$ [12].

**Methodology.** Building on advances in dataset distillation, we now introduce three variants of CLPBA that differ in the distance function (Equation 5) and the space of optimization:

- **Parameter Matching** (PM): Inspired by Trajectory Matching [7], PM attack aims to craft perturbations that encourage the victim model trained on the poison samples to have the same training trajectory as the one trained on the dirty-label trigger dataset. Let $\boldsymbol{\theta}_t(\boldsymbol{\delta})$ be the attacker model after $t$ steps of gradient descent on the poison samples. We introduce $\boldsymbol{\theta}_t^*$ as the **backdoor expert model**, initialized from $\boldsymbol{\theta}_t(\boldsymbol{\delta})$, that is trained $m$ steps on dirty-label trigger datasets. For the victim model to follow the trajectory of **backdoor expert model**, this attack minimizes:

$$\mathcal{A}_{\text{PM}} = \frac{\big\|\boldsymbol{\theta}_{t+m}^* - \boldsymbol{\theta}_{t+1}(\boldsymbol{\delta})\big\|_2^2}{\big\|\boldsymbol{\theta}_{t+m}^* - \boldsymbol{\theta}_t^*\big\|_2^2} \tag{7}$$

Specifically, $m > 1$ indicates that one gradient step on the poison dataset matches a long-range training trajectory ($m$ steps) on the dirty-label dataset of the **backdoor expert model**.

- **Gradient Matching** (GM): Instead of directly minimizing the distance $\boldsymbol{\theta}(\boldsymbol{\delta})$ and $\boldsymbol{\theta}^*$, which can be challenging in a high-dimensional parameter space with many local minima, GM attack, inspired by [54], minimizes the distance between the gradient updates of the attacker model trained on the poison samples and dirty-label datasets:

$$\mathcal{A}_{\text{GM}} = 1 - \frac{\Big\langle \nabla_{\boldsymbol{\theta}}\mathcal{L}^{P_t(\boldsymbol{\delta})}\left(\boldsymbol{\theta}(\boldsymbol{\delta})\right), \ \nabla_{\boldsymbol{\theta}}\mathcal{L}^{\widetilde{D}_s^p}\left(\boldsymbol{\theta}^*\right) \Big\rangle}{\left\|\nabla_{\boldsymbol{\theta}}\mathcal{L}^{P_t(\boldsymbol{\delta})}\left(\boldsymbol{\theta}(\boldsymbol{\delta})\right)\right\|_2 \left\|\nabla_{\boldsymbol{\theta}}\mathcal{L}^{\widetilde{D}_s^p}\left(\boldsymbol{\theta}^*\right)\right\|_2} \tag{8}$$

- **Feature Matching** (FM): GM and PM require solving a computationally expensive bi-level optimization problem. FM attack, inspired by [53], mitigates this by minimizing an empirical

estimate of the Maximum Mean Discrepancy (MMD) between the poisoned samples $P_t(\boldsymbol{\delta})$ and the source trigger distribution $\widetilde{\mathcal{D}}_s$ in a low-dimensional embedding space (i.e., the output of a feature extractor $f$ in a deep neural network). The empirical MMD is defined as:

$$\mathcal{A}_{\text{FM}} = \left\| \frac{1}{|\widetilde{D}_s|} \sum_{i=1}^{|\widetilde{D}_s|} f(\widetilde{\boldsymbol{x}}_i) - \frac{1}{|P_t|} \sum_{j=1}^{|P_t|} f(\boldsymbol{x}_j + \boldsymbol{\delta}_j) \right\|_2^2 \tag{9}$$

## 3.4 Enhancements for CLPBA

**Minimize approximation error.** We find that plain adaptation of data distillation methods to the CLPBA setting yields suboptimal performance due to the inherent approximation error between the attacker model used for crafting poisons and the victim model that is trained on the poison dataset. This gap arises from training randomness and differences in hyperparameters (e.g., batch size, learning rate). To reduce this mismatch, we employ three alignment techniques:

- **Iterative Re-training.** Since the poisoned model parameters $\boldsymbol{\theta}(\boldsymbol{\delta})$ depend on perturbations $\boldsymbol{\delta}$, which are dynamically updated during poison crafting with a fixed $\boldsymbol{\theta}$, it is necessary to iteratively retrain $\boldsymbol{\theta}(\boldsymbol{\delta})$ on perturbed dataset with the latest $\boldsymbol{\delta}$ after every $K$ optimization steps.

- **Trajectory Alignment.** Instead of using $\boldsymbol{\theta}$ of only the last training iteration to update perturbations, we keep a buffer $B = \{\theta_0, \theta_k, \theta_{2k}, \ldots\}$ to record the trajectory of the attacker model trained on the poison dataset. At each step, the attack will sample a $\boldsymbol{\theta}$ from $B$ to optimize perturbations.

- **Model Ensembling.** Following prior works [37, 1], we also employ an ensemble of models to craft poisons. Specifically, at each iteration, we averaged the gradients of the perturbations computed across all models before applying the update. We observed that this strategy reduces the variance in ASRs between random seeds of victim model training, increasing the transferability of the attack.

**Carlini-Wagner (CW) loss for GM attack.** Instead of using the standard cross-entropy objective to compute adversarial gradient $\nabla_{\boldsymbol{\theta}} \mathcal{L}^{P_t(\boldsymbol{\delta})}$, we use CW loss [6], which encourages high-confidence misclassification of trigger source-class samples:

$$\text{CW}(\boldsymbol{x}) = \max(F(\boldsymbol{x})_s - F(\boldsymbol{x})_t, -k), \quad \forall \boldsymbol{x} \in \widetilde{D}_s$$

where $k$ controls the desired misclassification confidence. CW loss empirically performs better than cross-entropy for GM attack, likely because it incorporates information of source-class logit in the gradient signal. While CW can also be adapted to PM attack to train backdoor experts, it yields inferior performance due to training misalignment between the backdoor expert and the victim model.

**Perturbation constraint.** Following prior work [37, 35, 50], we constrain perturbations to improve the stealthiness of poisoned samples. Typically, this is enforced via Projected Gradient Descent (PGD), which projects each perturbation onto the set $C = \{\boldsymbol{\delta} : \|\boldsymbol{\delta}\|_\infty < \epsilon\}$ at every step, where $\epsilon$ denotes the maximum allowed perturbation per pixel. However, this hard projection often introduces high-frequency noise that is visually noticeable in facial images. To address this, we replace the projection step with a visual loss term that is jointly optimized with the attack objective.

$$L_{\text{visual}} = \min(\text{abs}(\boldsymbol{\delta}) - \epsilon, 0) + \text{UTV}(\boldsymbol{\delta}),$$

where the first term softly enforces the $\ell_\infty$ constraint, and the second term (Upwind Total Variation [9]) regularizes local gradients between neighboring pixels to suppress high-frequency artifacts. Utilizing visual loss improves the perceptual quality of poison samples while maintaining or even improving ASR. We study the visual loss in-depth in the Appendix F.

We note that these proposed backdoor enhancements can be combined seamlessly in the pipeline of poison crafting. We refer readers to Appendix E for the algorithm and implementation details.

## 3.5 Connection to Hidden-Trigger Backdoor Attacks.

Our proposed GM and FM attacks share similarities with Sleeper Agent (SA) [37] and HTBA [35], as they optimize perturbations in the gradient and feature spaces. Despite having the same negative cosine loss function as SA, our GM attack can be considered an enhanced variant of SA with the mentioned improvements. Meanwhile, our FM attack differs from HTBA in the choice of objective: whereas HTBA minimizes pairwise distances between poisoned samples and trigger samples, FM minimizes the Maximum Mean Discrepancy between the poison set and the trigger distribution.

## 4 Evaluation

**Data Collection.** We created a Facial Classification dataset in one month with 3,344 clean and 9,790 trigger images from 10 Asian volunteers using 7 physical triggers (see Figure 2). To increase racial diversity, we added 90 random classes from PubFig [24], totaling 12,675 clean images. For animal classification, we combined a Kaggle dataset [2] (45 mammal classes) with 330 clean and 1,406 trigger images (tennis ball, phone, book). Animal classification is more challenging due to variable trigger sizes and placements. Further details are in Appendix A.

**Training Settings.** We split the datasets 80:20 for training and testing. ResNet50 [18] pre-trained on VGGFace2 [5] is used for facial recognition, and ResNet18 pre-trained on ImageNet-1K [34] for animal classification. We use a learning rate of 0.001 for finetuning and 0.1 for linear probing, with a step scheduler. The models converge after 40 epochs, with 99% accuracy for facial recognition and 93% for animal classification.

**Attack Settings.** We use 50% of source-class trigger images for poisoning, and evaluate the Attack Success Rate (ASR) based on misclassifications as the target class. CLPBA attacks are optimized with signAdam and a cosine decay scheduler for 750 iterations. The perturbation budget $\epsilon$ is 16/255, and the poison ratio $\alpha$ is 10%. CLPBA is evaluated with "sunglasses" and "fake beard" triggers for facial recognition, and "tennis ball" and "phone" for animal classification, using fixed source-target class pairs for comparison.

Table 1: ASR (%) of CLPBA and Baseline methods. We fix $\alpha = 10\%$ and $\epsilon = 16/255$ for CLPBA and LC. For CLPBA, we use an ensemble of 3 models with 3× retraining every 750 iterations. For consistency, we craft all attacks with a hard $\ell_\infty$ constraint.

| Trigger | Setting | Baseline | | | | CLPBA | | |
|---|---|---|---|---|---|---|---|---|
| | | Naive | LC | Dirty-label-d | Dirty-label-p | PM | GM | FM |
| **(a) Facial recognition** on ResNet50. Poison rates: **0.29%** - 30 images (sunglasses), **0.26%** - 26 images (fake beard). | | | | | | | | |
| sunglasses | linear | $0.0 \pm 0.0$ | $1.7 \pm 1.1$ | $72.7 \pm 18.5$ | $99.3 \pm 0.4$ | $88.6 \pm 5.3$ | $95.2 \pm 3.3$ | $98.2 \pm 0.8$ |
| | full | $0.0 \pm 0.0$ | $0.1 \pm 0.2$ | $17.3 \pm 7.9$ | $99.5 \pm 0.2$ | $65.8 \pm 5.5$ | $99.1 \pm 0.7$ | $99.3 \pm 0.3$ |
| fake beard | linear | $0.0 \pm 0.0$ | $12.6 \pm 15.7$ | $85.7 \pm 10.5$ | $99.7 \pm 0.5$ | $100.0 \pm 0.0$ | $99.3 \pm 1.2$ | $100.0 \pm 0.0$ |
| | full | $0.0 \pm 0.0$ | $1.0 \pm 1.6$ | $59.5 \pm 5.5$ | $100.0 \pm 0.0$ | $99.8 \pm 0.4$ | $100.0 \pm 0.0$ | $100.0 \pm 0.0$ |
| **(b) Animal classification** on ResNet18. Poison rates: **0.23%** - 27 images (tennis ball), **0.24%** - 30 images (phone). | | | | | | | | |
| tennis | linear | $0.0 \pm 0.0$ | $0.5 \pm 0.3$ | $72.6 \pm 3.8$ | $89.9 \pm 0.6$ | $93.8 \pm 0.9$ | $95.1 \pm 0.3$ | $93.7 \pm 0.2$ |
| | full | $0.1 \pm 0.1$ | $0.9 \pm 0.5$ | $26.6 \pm 3.5$ | $73.0 \pm 3.9$ | $26.9 \pm 11.5$ | $75.3 \pm 4.9$ | $59.2 \pm 9.5$ |
| phone | linear | $0.0 \pm 0.0$ | $0.1 \pm 0.1$ | $35.0 \pm 3.2$ | $77.9 \pm 0.7$ | $84.7 \pm 2.7$ | $87.1 \pm 1.8$ | $87.7 \pm 0.9$ |
| | full | $0.0 \pm 0.0$ | $0.0 \pm 0.0$ | $1.2 \pm 0.7$ | $56.4 \pm 1.8$ | $2.2 \pm 0.7$ | $61.5 \pm 4.6$ | $32.2 \pm 12.5$ |

**Baseline comparison.** We compare CLPBA with four baselines: (1) **Naive** attack, where the attacker adds samples from $\widetilde{\mathcal{D}}_t$ to the target-class data; (2) **Dirty-label-p** attack, where mislabelled samples from $\widetilde{\mathcal{D}}_s$ are inserted into the target-class data; (3) **Dirty-label-d** is the standard digital attack that embeds $p$ (i.e., the digital image of the physical trigger) to training samples in $D_s$ and change their labels from $s$ to $t$; and (4) **Label-Consistent (LC)** attack [39], in which the attacker perturbs the samples so that the victim model fails to classify them, and then overlays $p$ onto the perturbed images to make it a dominant feature (see Appendix B). We adapt the Naive attack to Animal classification by embedding $p$ onto target-class samples, due to the lack of trigger images. To improve the transferability of attacks with a digital trigger, we map $p$ to the appropriate facial position in Facial recognition, while randomizing the trigger locations in Animal classification. We note that Narcissus [50] and COMBAT [21] are not suitable baselines since these methods optimize triggers that are both used during training and inference, while CLPBA predefines a natural physical trigger used for inference-time misclassification. For each attack, we run 3 trials to calculate the average and standard deviation of ASR on source-class trigger images.

### 4.1 Attack Performance

**Comparison with baselines (Table 1).** In the Facial recognition task, where the position and size of physical triggers remain static relative to human faces, **Dirty-label-p** naturally achieves high performances, and CLPBA maintains competitive results with FM reaching near-perfect ASRs across multiple configurations. Even in this easy attack setting, we can observe that **Dirty-label-d** fails for full-finetuning scenarios, which validates our hypothesis about the lack of generalizability of digital

backdoor attacks. In a more challenging task like Animal classification, where trigger appearance varies widely in location, shape, and size, **CLPBA consistently outperforms the Dirty-label-p baseline** across all configurations. For example, FM achieves an ASR improvement of 9.8% under linear-probing with phone trigger, while GM has a 5.1% increase under full-finetuning setting with phone trigger. Two other baselines (Naive, LC) fail in all settings, with most ASRs below 1%. We note that not all CLPBA variants have good performance, as PM has low ASRs for the full-finetuning setting of Animal classification; however, it still has higher ASRs than **Dirty-label-d** baseline. Overall, GM attack achieves the best performance out of all the evaluated methods.

**Analysis.** Interestingly, CLPBA attacks outperform Dirty-label attacks even with preserved ground-truth labels and constrained perturbations. We attribute the limited effectiveness of Dirty-label attacks to their memorization property, and the small number of dirty-label poisons cannot sufficiently cover the distribution of $\widetilde{D}_s$ for test-time samples. CLPBA's superiority over these baselines stems from learning generalizable backdoor features rather than plain memorization. In other words, **CLPBA embeds representative trigger features through optimized perturbations**, enabling robust performance across diverse physical conditions.

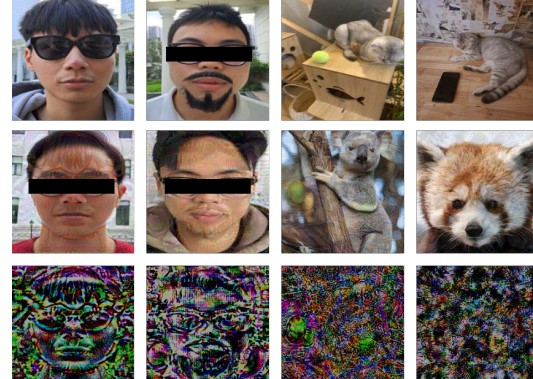

Figure 4: First row: sample in $\widetilde{D}_s$. Second row: Perturbed target-class samples. Third row: Scaled perturbations applied to target-class samples.

As visualized in Figure 4, we can observe the shape of sunglasses and real-beard triggers being constructed in perturbed images (columns 1-2), while multiple tennis ball features are embedded in the koala poison image (column 3).

**Comparison with hidden-trigger backdoor attacks (Figure 2).** Regarding gradient-space attacks, GM outperforms the SA attack by more than 10% for both triggers by integrating the proposed enhancement techniques (CW Loss + Trajectory Sampling + Visual Loss). Regarding feature-space attacks, FM surpasses HTBA by a substantial margin as HTBA remains ineffective with ASRs near zero.

Table 2: ASR (%) of CLPBA with backdoor enhancements and hidden-trigger baselines on ResNet18 (full-finetuning).

| Trigger | SA | GM (ours) | HTBA | FM (ours) |
|---|---|---|---|---|
| tennis | $62.9 \pm 7.1$ | $\mathbf{74.2 \pm 3.6}$ | $1.1 \pm 0.2$ | $\mathbf{57.5 \pm 2.9}$ |
| phone | $51.1 \pm 4.9$ | $\mathbf{65.5 \pm 2.1}$ | $0.1 \pm 0.1$ | $\mathbf{30.1 \pm 1.0}$ |

## 4.2 Ablation Study

Table 3: Ablation study on the animal classification task with ResNet18, full fine-tuning, and a tennis trigger ($\alpha = 0.1$, $\epsilon = 16$). ASR (%) is reported. The "Single" column shows the effect of each component in isolation, while the "Combine" column reports results with cumulative components. The highest ASR in each column is highlighted.

| | GM | | FM | |
|---|---|---|---|---|
| | Single | Combine | Single | Combine |
| Baseline | $22.3 \pm 5.3$ | | $14.9 \pm 2.5$ | |
| + CW Loss | $\mathbf{67.5 \pm 3.7}$ | | N/A | |
| + Retrain | $52.3 \pm 1.7$ | $60.9 \pm 7.4$ | $\mathbf{44.1 \pm 4.9}$ | |
| + Ensemble | $43.0 \pm 5.9$ | $\mathbf{82.3 \pm 1.7}$ | $12.6 \pm 5.3$ | $52.2 \pm 5.0$ |
| + TrajAlign | $32.5 \pm 5.3$ | $77.8 \pm 3.3$ | $28.0 \pm 6.8$ | $56.8 \pm 3.3$ |
| + Visual Loss | $46.7 \pm 6.1$ | $78.3 \pm 2.3$ | $17.6 \pm 6.3$ | $\mathbf{57.5 \pm 2.9}$ |

In Table 3, we measure ASR(%) improvement when adding a single enhancement and adding a combination of enhancement techniques. Compared to the baseline, where no technique is applied, the integration of the proposed enhancements improves GM attack and FM attack by a maximum of **60.0%** and **42.6%**. For the GM attack, every technique applied individually is shown to improve ASR significantly; CW Loss is the most notable with the increase of **45.2%**. We can observe that

Trajectory Alignment with other techniques does not increase the ASR of the GM attack over the combination of (CW Loss, Retraining, Ensembling). We believe that this is because we didn't set a sufficiently large number of attack iterations, which prevented the attack with Trajectory Alignment from converging optimally. On the other hand, for the FM attack, the combination of all backdoor enhancement techniques results in the highest ASR of 57.5%. Iterative Retraining is the most important enhancement for this attack, with an improvement of 28.2%.

# 5   Defending against CLPBA

We evaluated our Clean-Label Poisoning Backdoor Attack (CLPBA) against 15 representative defenses belonging to four families of defenses. Overall, CLPBA demonstrates significant robustness, evading most existing state-of-the-art defenses. We refer readers to Appendix G for a description of evaluated defenses and full experiment results. Below is the summary of our evaluation:

**Preprocessing-Based Defenses** [51, 48]. These defenses apply strong data augmentations to weaken triggers during training. We find strong augmentations, such as MixUp [51] and CutMix [48], are largely **ineffective** against CLPBA. While Noising and Denoising augmentations can partially mitigate the attack since they disrupt the perturbations applied on poison samples, their effectiveness is nullified by a simple **adaptive attack**, where the attacker applies the same augmentation during poison crafting.

**Filtering Defenses** [4, 31, 38, 17, 20]. Out of 5 evaluated filters, we only find Spectral Signature defenses (SS [38], SPECTRE [17]) can correctly filter poison samples with high True Positive Rate. This is perhaps not surprising since CLPBA's poison samples contain features of trigger distribution, separating them from the natural distribution of the target class in the feature space. However, the downsides of these defenses are high False Positive Rate, removing up to **30.4%** of clean samples to successfully weaken the attack.

**Firewall Defenses** [13, 19, 49, 45]. These defenses aim to block the inference of victim models on malicious inputs at test time. We find that CLPBA is highly effective against these defenses. We believe that such defenses are designed specifically for dirty-label backdoor attacks, preventing their application to the clean-label backdoor attacks.

**Backdoor Detection** [40, 28]. These defenses analyze the trained model to determine if it has been compromised, and reverse-engineer the triggers to purify the compromised model from the backdoor attack. We find Neural Cleanse (NC) [40] is **ineffective** against CLPBA, successfully identifying the backdoored class in only **2 out of 10 trials**. NC uses Anomaly Index to detect the target class and the associated trigger, with the assumption that the trigger should have an unusually smaller norm for the target class than for other classes. This assumption is clearly violated by CLPBA with the use of physical triggers that are subjected to physical variability. ABS [28] is another detection method that aims to detect malicious neurons related to the backdoor attack before synthesizing the backdoor trigger. This method is also **ineffective** against CLPBA since it consistently associating malicious neurons with incorrect target classes and creating poor-quality triggers with **0.0% ASR**.

**Backdoor Mitigation.** These defenses attempt to cleanse poisoned models using small sets of clean data. We find I-BAU [44] is **ineffective** against CLPBA, with this adversarial unlearning method barely impacting the attack by only reducing ASR from 97.7% to **93.3%**. However, NAD [25] is **highly effective**, successfully purging the backdoor through Neural Attention Distillation and reducing ASR from 97.7% to just **3.3%** without damaging clean data accuracy. While NAD can mitigate CLPBA, the impact on ACC may depend on the amount of clean samples that the defender has for finetuning. Furthermore, the dependence of clean dataset limits its applicability for scenarios where third-party Machine Learning services are responsible for training the models.

# 6   Conclusion

We introduce Clean-Label Physical Backdoor Attacks (CLPBA), a new paradigm for physical backdoor poisoning that eliminates the need for label manipulation and trigger injection. Formulating the attack as a dataset distillation problem, we developed three CLPBA variants and introduced backdoor enhancement techniques that together craft highly effective and stealthy poison samples that can even surpass Dirty-label attacks in hard scenarios where backdoor generalizability is required.

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

# A  Dataset Collection & Pre-processing

## A.1  Ethics & Data Protection

**IRB approval.** Before conducting our study, we submitted a "Human Subjects Research – Expedited Review Form" to our country's Institutional Review Board (IRB). Our study received approval from the chairman of the institutional ethical review board under decisions *No 24/2016/QD-VINMEC*, *No 23/2016/QD-VINMEC*, and *No 77/2021/QD-VINMEC*. We prepared a consent form beforehand to ensure transparency in the procedure of dataset collection. All 10 volunteers in our dataset provided **explicit written consent** for us to collect the dataset and use the images for research purposes, including permission to use the captured images in the research paper.

**Dataset protection.** In adherence to strict ethical standards and privacy considerations related to the sensitive nature of the human face dataset, our research follows a comprehensive protocol to protect the privacy and confidentiality of the collected data. To safeguard the data, all images are securely stored on a protected server, with access restricted solely to the authors for research purposes. Additionally, all images in the paper are partially obscured to ensure that the identities of our volunteers are not exposed.

## A.2  Dataset Collection

Due to the lack of publicly available datasets to study physical backdoor attacks, we collect a facial recognition dataset with 10 identities that contains 3,344 clean images and 9,790 trigger images of 7 physical triggers. Sample images of identities are given in Figure 5. To reflect real-world conditions, the dataset was constructed in 1 month so that the images could be captured in various indoor/outdoor settings, under varying weather conditions, and with diverse shooting angles and distances. All photos are RGB and of size (224,224,3), taken using a Samsung Galaxy A53 and Samsung Galaxy S21 FE. To enhance the racial diversity of our dataset, we merge the collected dataset with 90 classes of the PubFig dataset [24], resulting in a total of 12,675 clean images.

In our dataset, we choose triggers based on 3 criteria:

- **Stealthiness**: Does the trigger look natural on a human face?

- **Size**: How big is the trigger?

- **Location**: Is the trigger on-face or off-face?

With these criteria, we select 7 triggers, as shown in Table 4.

Table 4: Our assessment of chosen physical triggers

| Trigger | Stealthiness | | On-Face | | Size | | |
|---|---|---|---|---|---|---|---|
| | Yes | No | Yes | No | Small | Medium | Big |
| Earrings | ✓ | | | ✓ | ✓ | | |
| Fake Beard | ✓ | | ✓ | | | ✓ | |
| Sticker | | ✓ | ✓ | | ✓ | | |
| Facemask | | ✓ | ✓ | | | | ✓ |
| Hat | ✓ | | | ✓ | | ✓ | |
| Sunglasses | | ✓ | ✓ | | | | ✓ |
| Headband | ✓ | | ✓ | | | ✓ | |

## A.3  Dataset Preprocessing

After collecting the images, we utilize a pre-trained MTCNN [52] (Multi-task Cascaded Convolutional Networks) model to detect and crop the face area. This preprocessing step ensures that the face is the focal point of each image, effectively removing any background noise. The cropped face regions are then resized to a standard dimension of $224 \times 224$ pixels.

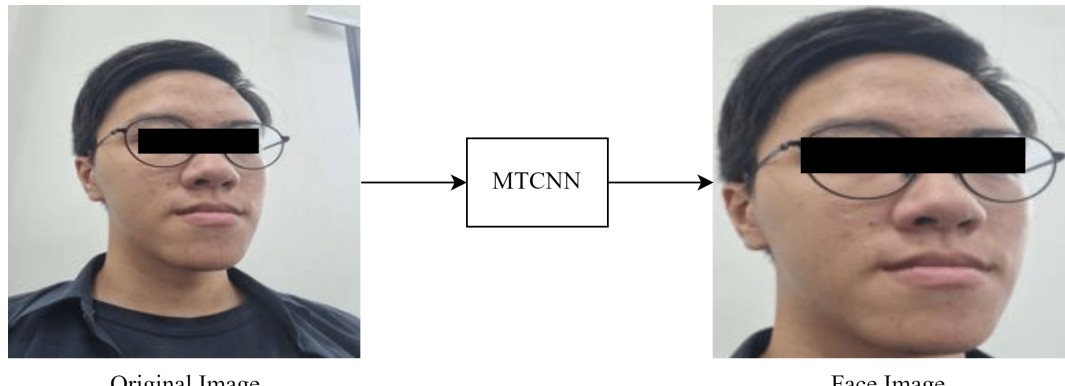

Figure 5: Visualization of the face detection process. **Left**: Original image. **Right**: Processed image with face area cropped and resized to $224 \times 224$ pixels.

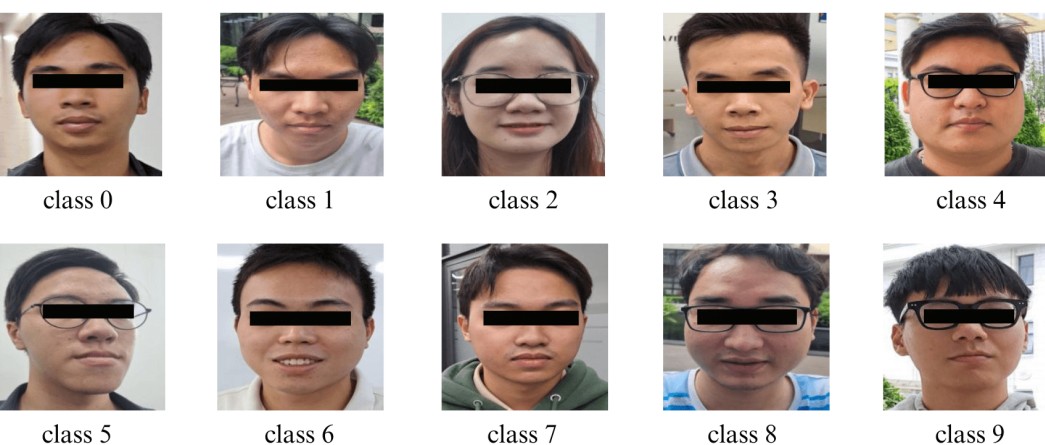

| class 0 | class 1 | class 2 | class 3 | class 4 |
| class 5 | class 6 | class 7 | class 8 | class 9 |

Figure 6: Example images of the 10 volunteers, representing the first 10 classes in our facial recognition dataset.

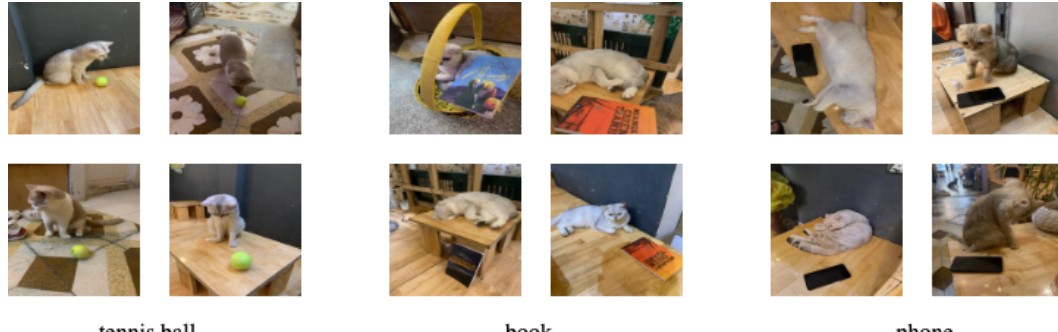

tennis ball                    book                    phone

Figure 7: Physical triggers of the animal classification dataset.

### A.4 Animal Classification Dataset

Besides facial recognition, we also evaluate CLPBA on animal classification. We collected 1,670 cat images (264 clean images + 1406 trigger images) of three physical triggers: tennis balls, mobile phones, and books. The trigger-free cat images are resized to $224 \times 224$ and then concatenated to an existing animal classification dataset on Kaggle [2] to create an animal classification dataset with a total of 14,091 clean images of 46 species. Visualization of trigger images for this dataset is given in Figure 7.

## B    Comparison between CLPBA and baselines

**Label-Consistent Attack.** Label-Consistent (LC) attack [39] works by perturbing the poisoned samples with adversarial noise $\delta$ to make the salient features of the samples harder to learn (by ascending the cross-entropy loss on these samples) before injecting the **digital trigger** $p$ on perturbed samples, forcing the model's classification to depend on $p$ for target-class classification.

$$\boldsymbol{x} \leftarrow \boldsymbol{x} + \underset{\|\boldsymbol{\delta}\|_\infty \leq \epsilon}{\arg\max} \mathcal{L}(F_{\boldsymbol{\theta}}(\boldsymbol{x}), t) + p \ , \forall \boldsymbol{x} \in D_t^{\text{pois}}$$

To adapt LC to our setting, we extract the digital pattern of the physical trigger to embed on poison images (Figure 8).

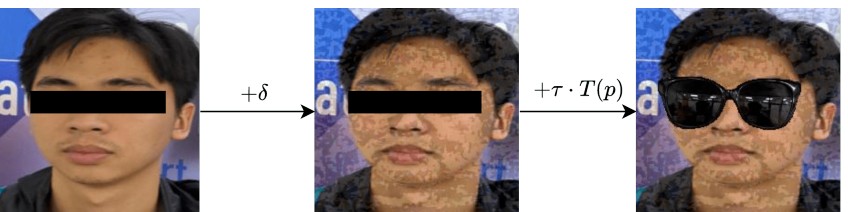

Figure 8: Procedure of LC attack in Facial Recognition. We add the adversarial perturbations to the poison instance before inserting the trigger into the appropriate facial area.

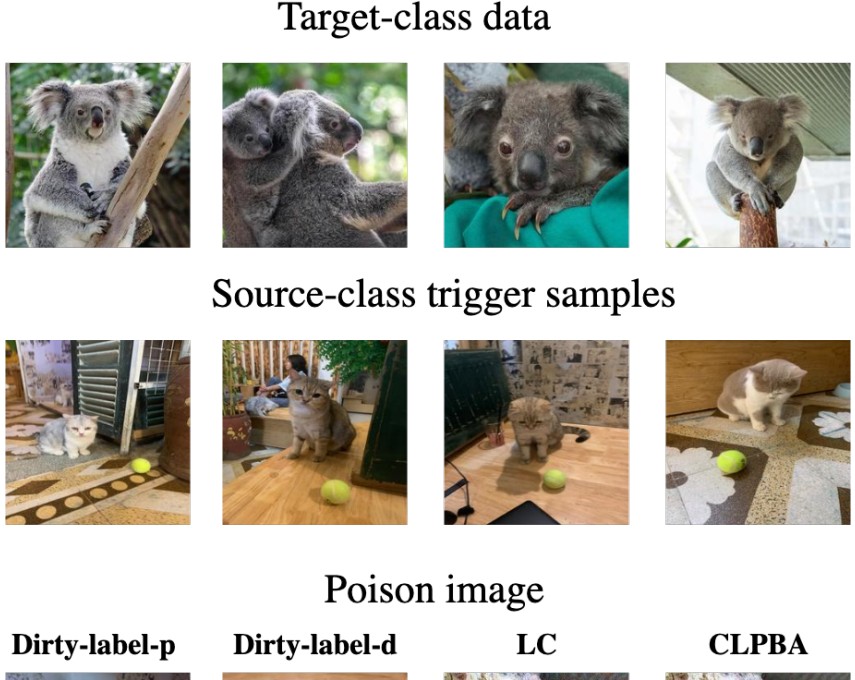

Figure 9:  Visualization of baseline attacks and CLPBA with tennis trigger, and cat-koala is the source-target class pair. Red label means that the sample has its label changed to the target class, while the Green label preserves ground-truth labelling.

**Dirty-label-p Attack.** This is a strong baseline that involves the attacker injecting dirty-label trigger samples from the source class to the training dataset and changing their ground-truth labels to the

target class. Since the injected samples are drawn directly from the trigger distribution, a sufficiently high poison ratio will ensure that the physical backdoor attack is successful.

**Dirty-label-d Attack.** This is an adapted version of digital backdoor attacks, where the attacker embeds the digital trigger into the source-class images and flips their labels to the target class.

**Naive Attack.** A naive clean-label attack where the attacker injects trigger images from the target class into the training dataset to create a connection between the physical trigger and target-class feature space. This attack assumes that the attacker has trigger samples from the target class.

Visualizations of the poison image for each of these baselines are given in Figure 9.

# C   Detailed Experiment Settings

Table 5: Victim Hyperparameters of CNN architectures

| Hyperparameter | Value |
|---|---|
| Optimizer | SGD [33] |
| Full-finetuning lr | 0.001 |
| Linear-probing lr | 0.1 |
| Lr scheduler | Drop by 90% every 10 epochs |
| Decay rate | 5e-4 |
| Batch size | 64 |
| Training epochs | 40 |

## C.1   Computational Resources.

All the experiments are conducted on the two servers. The first one with 7 RTX 3090 Ti 24 GB GPUS, and the second one has 6 RTX A5000 24 GB GPUS. The code is implemented in PyTorch. We develop the codebase based on the previous works [37, 14].

## C.2   Training hyperparameters.

We summarize key hyperparameters of the victim model training for our main table results in Table 5. We use the same set of hyperparameters across CNN architectures, while for Vision Transformers, we set a lower learning rate of 0.0001 for full-finetuning and 0.001 for linear-probing.

## C.3   Evaluation Metrics.

To evaluate the performance of CLPBA, we adopt two standard metrics for backdoor attacks:

• **Attack success rate (ASR) (%)**: The proportion of examples in a trigger dataset of the source-class that the model misclassifies as the target class at inference time.

• **Accuracy (ACC) (%)**: The model's prediction accuracy on clean, ordinary test-time data.

Table 6: ACC of the victim model before and after GM attack under full-finetuning scenario.

| | Pre-Attack | Post-Attack |
|---|---|---|
| ResNet50 (Face recogntion) | 99.7 | $99.8 \pm 0.0$ |
| ResNet18 (Animal classification) | 93.6 | $93.8 \pm 0.1$ |

Since we observe that ACC is only minimally affected by the attacks or even increased after the attack (Table 6), we omit this metric in our experiments. We note that our experiments use a low poison ratio of around 0.2% to 0.3% poison ratio over the whole training set, which explains why ACC is not affected in most cases.

## D  Proof for Proposition 1

The proof of Proposition 1 is derived based on the Implicit Function Theorem [23]:

*Proof.* By the chain rule, the gradient of $\mathcal{A} = D(\boldsymbol{\theta}(\boldsymbol{\delta}))$ is:

$$\nabla_{\boldsymbol{\delta}} \mathcal{A} = \left( \frac{\partial \boldsymbol{\theta}(\boldsymbol{\delta})}{\partial \boldsymbol{\delta}} \right)^{\top} \nabla_{\boldsymbol{\theta}} D$$

While $\nabla_{\boldsymbol{\theta}} D$ is trivial to compute, we focus on the implicit gradient $\frac{\partial \boldsymbol{\theta}}{\partial \boldsymbol{\delta}}$. Since $\boldsymbol{\theta}(\boldsymbol{\delta})$ is the minimizer of $\mathcal{L}^{P_t(\boldsymbol{\theta})}$ by construction, we can define $\boldsymbol{\theta}(\boldsymbol{\delta})$ implicitly by the optimality condition:

$$\nabla_{\boldsymbol{\theta}} \, \mathcal{L}^{P_t(\boldsymbol{\delta})}(\boldsymbol{\theta}) \, \Big|_{\boldsymbol{\theta} = \boldsymbol{\theta}(\boldsymbol{\delta})} = \mathbf{0}$$

We differentiate the equation $\nabla_{\boldsymbol{\theta}} \mathcal{L}(\boldsymbol{\theta}(\boldsymbol{\delta})) = \mathbf{0}$ with respect to $\boldsymbol{\delta}$ using the total derivative and applying the chain rule:

$$\frac{d}{d\boldsymbol{\delta}} \left[ \nabla_{\boldsymbol{\theta}} \mathcal{L}^{P_t(\boldsymbol{\delta})} \right] = \nabla_{\boldsymbol{\delta}} \nabla_{\boldsymbol{\theta}} \mathcal{L}^{P_t(\boldsymbol{\delta})} + \left( \nabla_{\boldsymbol{\theta}}^2 \mathcal{L}^{P_t(\boldsymbol{\delta})} \right) \frac{\partial \boldsymbol{\theta}}{\partial \boldsymbol{\delta}} = \mathbf{0}$$

Using the definitions for $\mathbf{G}$ and $\mathbf{H}$, this is $\mathbf{G} + \mathbf{H}\frac{\partial \boldsymbol{\theta}}{\partial \boldsymbol{\delta}} = \mathbf{0}$. We solve for the Jacobian:

$$\frac{\partial \boldsymbol{\theta}}{\partial \boldsymbol{\delta}} = -\mathbf{H}^{-1}\mathbf{G}$$

Substituting this into the chain rule expression:

$$\nabla_{\boldsymbol{\delta}} \mathcal{A} = \left( -\mathbf{H}^{-1}\mathbf{G} \right)^{\top} \nabla_{\boldsymbol{\theta}} D = -\mathbf{G}^{\top} (\mathbf{H}^{-1})^{\top} \nabla_{\boldsymbol{\theta}} D$$

Since the Hessian $\mathbf{H}$ and its inverse are symmetric, $(\mathbf{H}^{-1})^{\top} = \mathbf{H}^{-1}$, we obtain the result:

$$\nabla_{\boldsymbol{\delta}} \mathcal{A} = -\mathbf{G}^{\top} \mathbf{H}^{-1} \nabla_{\boldsymbol{\theta}} D$$

This completes the proof. $\qquad\square$

**Discussion.** This result is a direct application of the implicit function theorem to the bilevel structure of CLPBA. It highlights three key quantities: (i) $\mathbf{G}$ transfers the effect of pixel-level perturbations $\boldsymbol{\delta}$ onto the model parameters through the training loss, (ii) $\mathbf{H}^{-1}$ measures the local curvature of that loss, and (iii) $\nabla_{\boldsymbol{\theta}} D$ steers the parameters toward the dirty-label optimum $\boldsymbol{\theta}^*$. Since $H^{-1}$ is computationally expensive, and the exact solution of $\boldsymbol{\theta}(\boldsymbol{\delta})$ is intractable for large networks; practical attackers use *unrolled optimisation*, i.e. back-propagating $\boldsymbol{\theta}$ through a finite inner loop of $K$ gradient-descent steps training on $P_t(\boldsymbol{\delta})$, as proposed by [12].

## E  Algorithm and Implementation Details

### E.1  Algorithm and Implementation.

The full algorithm of CLPBA, with the proposed enhancements, is given in Algorithm 1. First, the attacker initializes and trains the attacker models, storing the model checkpoints in the buffer $B$ (Lines 1-3). In our codebase, however, the buffer will store algorithm-specific inputs to avoid repeated computations in the inner loops:

- **GM:** The buffer stores adversarial gradients of models at different timesteps.

- **FM:** The buffer stores the weights of models at different timesteps.

- **PM:** The buffer stores a pair of (starting parameters, target parameters), where the starting parameters are the parameters that train normally on the training data $D$, and the target parameters are the parameters of the expert backdoor model that have been fine-tuned on dirty-label backdoor data.

---

**Algorithm 1** CLPBA poison crafting procedure

---

**Input**: Training dataset $D$, source triggerset $\widetilde{D}_s$.

**Parameter**: Perturbation budget $\epsilon$, poison budget $\alpha$, retrain factor $R$, optimization step $K$, learning rate for updating perturbations $\eta$, number of models in an ensemble $M$, weight of the visual loss $\lambda_{\text{visual}}$.

**Output**: The set of poison samples: $P_t(\boldsymbol{\delta}) = \left\{ (\boldsymbol{x}_i + \boldsymbol{\delta}_i, t) \mid (\boldsymbol{x}_i, t) \in D_t^{\text{pois}} \right\}$, where $D_t^{\text{pois}} \subset D_t \subset D$ contain the target-class samples that the attacker can perturb.

1: Initialize the attacker model $\mathcal{F}$ as an ensemble of models: $\mathcal{F} = \{F_{\boldsymbol{\theta}^{(1)}}^{(1)}, F_{\boldsymbol{\theta}^{(2)}}^{(2)}, \ldots F_{\boldsymbol{\theta}^{(M)}}^{(M)}\}$.
2: Initialize a buffer $B$ to store the trajectory of every model $\mathcal{F}$.
3: Train each of the models in $\mathcal{F}$ with the training data $D$ and fill up $B$ with checkpoints for every timestep $k$: $B = \{(\boldsymbol{\theta}_k^{(1)}, \ldots \boldsymbol{\theta}_k^{(m)}), (\boldsymbol{\theta}_{2k}^{(1)}, \ldots \boldsymbol{\theta}_{2k}^{(m)}), \ldots\}$.
4: Under the poison budget $\alpha$, select $N_p$ samples from $D_t$ to create $D_t^{\text{pois}}$.
5: Initialize $\boldsymbol{\delta} = \{\boldsymbol{\delta}_1, \ldots \boldsymbol{\delta}_{N_p}\}$ as perturbations for $D_t^{\text{pois}}$.
6: **for** $r = 1, 2, \ldots, R$ **do**
7:    **for** $t = 1, 2, \ldots, T$ **do**
8:       Sample a set of weights $\boldsymbol{\theta} \sim B$ representing a specfic timestep.
9:       Sample a batch $\widetilde{b}_s \sim \widetilde{D}_s$ and a batch $b_t \sim P_t(\boldsymbol{\delta})$.
10:      Compute attacker objective: $L_{adv} \leftarrow \mathcal{A}(\boldsymbol{\theta}, b_t, \widetilde{b}_s; \ \boldsymbol{\delta})$
11:      **if** visual loss is used **then**
12:        Compute visual loss:

$$L_{\text{visual}} = \sum_i \max\left(|\delta_i| - \epsilon, \, 0\right) + \sum_{i,j} \left[ (\delta_{i+1,j} - \delta_{i,j})^2 + (\delta_{i,j+1} - \delta_{i,j})^2 \right]$$

13:        Compute the gradient $\nabla_{\boldsymbol{\delta}} (\mathcal{A} + \lambda_{\text{visual}} L_{\text{visual}})$ and update $\boldsymbol{\delta}$ with signed Adam and a learning rate $\eta$.
14:      **else**
15:        Compute the gradient $\nabla_{\boldsymbol{\delta}} (\mathcal{A})$ and update $\boldsymbol{\delta}$ with signed Adam and a learning rate $\eta$.
16:        Project $\boldsymbol{\delta}$ to constraint set $C = \{ \boldsymbol{\delta} : ||\boldsymbol{\delta}_i||_\infty \leq \epsilon, \forall i \}$.
17:      **end if**
18:      Ensure that every sample in $P_t(\boldsymbol{\delta})$ stays within the range $[0, 1]$ after $\boldsymbol{\delta}$ is updated.
19:    **end for**
20:    Reinitialize the buffer $B$.
21:    Retrain the attacker model $\mathcal{F}$ on the poison dataset $D^p = \left( D \setminus D_t^{\text{pois}} \right) \cup P_t(\boldsymbol{\delta})$ and fill up the buffer $B$.
22: **end for**
23: **return** $P_t(\boldsymbol{\delta})$.

---

After initializing the buffer, the attacker proceeds to optimize perturbations $\boldsymbol{\delta}$ that are to be added to target-class samples $D_t^{\text{pois}}$. **Inner-loop** (Lines 6-17): At each optimization step, the attacker samples a batch of source-class trigger samples $\widetilde{b}_s$ and a batch of target-class poison samples $b_t$ to optimize $\boldsymbol{\delta}$ with the attacker objective (as defined in the Methodology section of the main paper). If visual loss is used (Lines 10-12), the attackers compute $L_{adv}$ as the sum of soft $\ell_\infty$ penalty (first term) and Upwind Total Variation (second term) [9]. **Outer-loop** (Lines 19-20): After $K$ optimization steps, the attacker reinitializes the buffer $B$ and re-trains the attacker model on the updated poison dataset to fill up the buffer.

As observed in Algorithm 1, our proposed backdoor enhancement components are used in different parts of the algorithm, and thus can be combined naturally. During **Iterative Re-training**, the buffer stores the checkpoints for **Trajectory Alignment** to minimize approximation error between attacker and victim models. **Visual loss** is optimized along with the attacker's objective to improve the perceptual quality of perturbations. **CW Loss** is used during re-training steps to store the adversarial gradients for the GM attack.

**Discussion.** Three points are worth mentioning: **(1)** The three attacks represent three spaces of optimization for the perturbations: parameter space, gradient space, and feature space. PM can be

thought of as an extension of GM, where one gradient step on $P_t(\boldsymbol{\delta})$ matches $m$ gradient steps on $\widetilde{D}_s^p$. However, we find that the performance of PM is often inferior to GM (Table 1 in the main paper). The reason is that training the expert model on $\widetilde{D}_s^p$ causes its training trajectory to drift farther away from the trajectory of the victim model, and thus optimizing $\mathcal{A}_{\text{PM}}$ cannot reliably approximate the adversarial learning dynamics of the victim model on poisoned training data. **(2)** Compared to PM and GM, FM is more efficient since it does not involve solving inner-loop optimization with higher-order gradients $\nabla_{\boldsymbol{\delta}}\nabla_{\boldsymbol{\theta}}\mathcal{L}^{P_t(\boldsymbol{\delta})}$. **(3)** Our formulation of CLPBA as a data distillation problem allows for a more general case of data-poisoning attacks where information of an arbitrary source distribution $\mathcal{D}_s$, which may not necessarily represent a class in the training set, is embedded into the target class for test-time misclassification. We leave this direction for future work.

## E.2 Hyperparameters for CLPBA.

We discuss important hyperparameters for CLPBA and its influence on attack performance and stealthiness:

$\epsilon = 8$ $\qquad\qquad$ $\epsilon = 16$ $\qquad\qquad$ $\epsilon = 32$ $\qquad\qquad$ $\epsilon = 64$

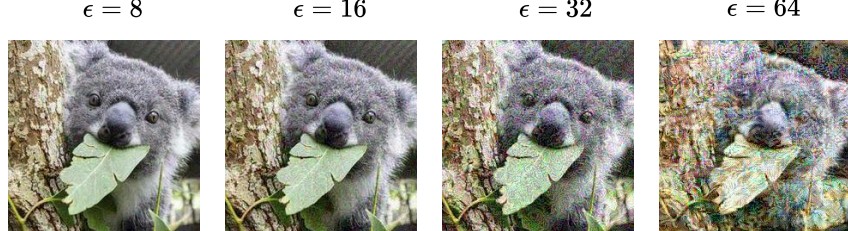

Figure 10: Visualization of perturbed "koala" images under GM attack with $\ell_\infty$ constraint.

- **Poison budget** $\alpha$: This determines the attacker's capability as it decides the number of target-class samples that the attacker can poison. In practice, with full access to the training data, the attacker can craft poisons with a high poison budget since the hidden-trigger and clean-label characteristics of CLPBA make the perturbed samples harder to detect via human inspection or automated filtering. Generally, higher $\alpha$ increases attack influence on victim models, allowing the attacker's objective to converge to a lower value.

- **Perturbation budget** $\epsilon$: Setting a high value of $\epsilon$ also improves attack convergence; however, it may compromise the perceptual quality of perturbed images. We find that a value of 8 to 16 (Figure 11a) is an appropriate range for our facial recognition task that balances between performance and stealthiness, while for the animal classification task, where there is more natural background, we can set a higher value of 16 to 32 (Figure 10 without greatly impacting the visual quality of poison images.

- **The weight for the visual loss** $\lambda_{\text{visual}}$ balances between attacker objective and visual stealth of poison samples. We set this to 1 across all configurations.

- **Number of inner-loop optimization steps** $K$: We find that a higher value of $K$ is beneficial to all of the attacks, since it allows for better convergence of $\mathcal{A}$. We generally set $K$ to be between 250 to 750 steps.

- **Retrain factor** $R$: Similar to $K$, a higher retrain factor improves attack success as it better aligns the attack model with the victim model. However, the effect saturates as $R$ increases. We note that setting a high $R$ results in a long running time since it involves re-training on the full poisoned dataset. We set $R$ to be between 1 to 5.

- **Learning rate to update perturbations** $\eta$: This parameter can be tuned to improve the convergence of the attack. In our experiments, we set $\eta$ to 0.1 for GM & PM and 0.01 for FM across all settings.

- Number of **backdoor training steps** $m$ for PM attack: Since PM naturally suffers from higher approximation error compared to GM and FM, since the backdoor expert model is fine-tuned on a dirty-label backdoor dataset, we set $m = 1$ to reduce the misalignment.

- **Batch size** for inner-loop optimization: We observe that a larger batch size for sampling from the poison set leads to a more effective attack since the adversarial loss can converge to a lower value.

 **F    Analysis of the Visual Loss.**

$\epsilon = 8$    $\epsilon = 16$    $\epsilon = 32$    $\epsilon = 64$

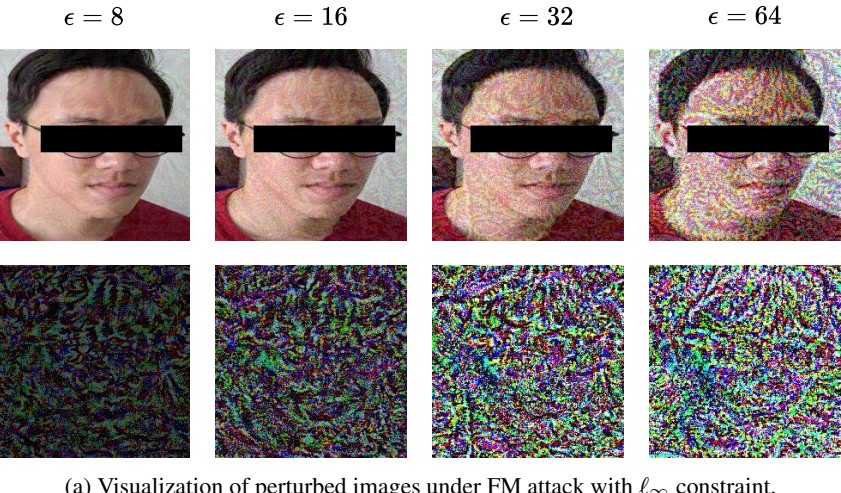

(a) Visualization of perturbed images under FM attack with $\ell_\infty$ constraint.

$\epsilon = 8$    $\epsilon = 16$    $\epsilon = 32$    $\epsilon = 64$

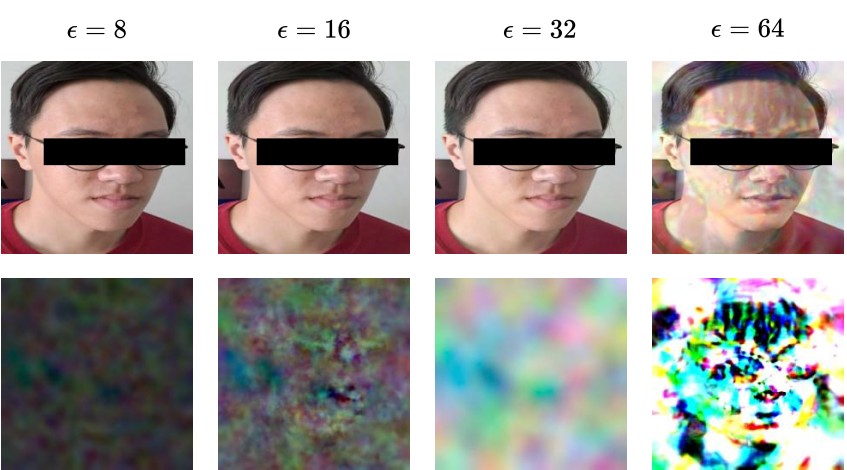

(b) Visualization of perturbed images under FM attack with visual loss.

Figure 11: Comparison of perturbed images under different FM attack constraints.

Our study reveals a nice complement of the Upwind Total Variation (UTV) term to the soft $\ell_\infty$ penalty. When using only the soft $\ell_\infty$ penalty, we observe that the attacker's objective dominates the penalty term during the optimization, causing the perturbations to grow quickly. This not only degrades the visual quality of the poison samples but also causes instability to the optimization process, as observed in fluctuations of the attacker's objective. We have also tested with $\ell_2$ regularization; however, this regularization tends to penalize the perturbation norm too heavily, which causes difficulty for the optimization process. UTV is a lighter regularization compared to $\ell_2$: Instead of penalizing pixel-level values, it penalizes the norm of gradients between neighbouring pixels, ensuring a smoother transition between a pixel to its neighbours. We analyze two nice properties of CLPBA with the visual loss: improved stealthiness of poison images and improved convergence of attacker objective.

**Visual Loss improves stealthiness.** We demonstrate the comparision between images perturbed with original $\ell_\infty$ constraint and images perturbed with the proposed visual loss in Figure 11. We observe that when $\epsilon$ is 16, we start seeing visible artifacts on face image with $\ell_\infty$ constraint. This effect is more notable with $\epsilon = 32$ and $\epsilon = 64$. On the other hand, using the visual loss effectively smoothen the perturbations, and preserves the perceptual quality of poison images. As can be observed in Figure 11b, even at a high value of $\epsilon = 32$, **there is no visible difference between poison image and original image**. We also record the Peak Signal-to-Noise Ratio (PSNR) in dB, a popular metric to evaluate the quality of corrupted images with. Higher values of PSNR indicate better image quality.

As shown in Table 8, the visual loss consistently achieves higher PSNR across all $\epsilon$, indicating the better stealthiness of perturbed samples.

**Visual Loss improves effectiveness.** We observe that the visual loss helps improve ASR over the standard $\ell_\infty$ constraint since it enables the attacker's objective to converge to a smaller value. Since the visual loss has a larger space of optimization compared to $\ell_\infty$, visual loss would benefit from a higher number of optimization steps. As can be seen in Table 7, both $\ell_\infty$ constraint and visual loss benefit from higher numbers of optimization steps, and visual loss outperforms $\ell_\infty$ constraint in 3 out of 4 tests. Notably, when $T = 750$, using the visual loss increases ASR (%) by **17.4%**.

Table 7: Comparison of GM attacks with $\ell_\infty$ constraint and visual loss under different numbers of optimization steps.

|  | $T = 250$ | $T = 500$ | $T = 750$ | $T = 1000$ |
|---|---|---|---|---|
| $\ell_\infty$ constraint | $32.5 \pm 1.6$ | $48.1 \pm 2.8$ | $42.8 \pm 1.4$ | $47.5 \pm 1.3$ |
| Visual loss | $44.6 \pm 1.2$ | $49.1 \pm 0.8$ | $60.2 \pm 2.1$ | $44.8 \pm 2.1$ |

Table 8: Comparison of Peak Signal-to-Noise Ratio (PSNR) (dB) under different perturbation budgets ($\epsilon$). Higher PSNR values indicate better perceptual image quality.

|  | $\epsilon = 8$ | $\epsilon = 16$ | $\epsilon = 32$ | $\epsilon = 64$ |
|---|---|---|---|---|
| $\ell_\infty$ constraint | 31.7 | 26.2 | 20.5 | 15.0 |
| Visual loss | 33.3 | 28.4 | 23.2 | 18.2 |

# G  Evaluation of Defenses

To defend against backdoor attacks in DNNs, defenses of different categories have been proposed. We summarize the families of defenses that we evaluate CLPBA against:

- **Preprocessing-based defenses**: These defenses aim to weaken embedded triggers by pre-processing the training data. Strong data augmentations (e.g., MixUp, CutMix) have been shown to improve the robustness of model training with poisoned data [51, 48]. Noise-based augmentation (e.g., Gaussian noising/denoising) has also been shown to be effective against perturbation-based attacks. Thus, we evaluate CLPBA against MixUp and CutMix augmentations, together with Gaussian noising/denoising.
- **Backdoor detection defenses** focus on detecting whether the model has been backdoored or not.
- **Filtering-based defenses** aim to filter poison samples during traing stage.
- **Firewall defenses** aim to safeguard to model from making inferences on suspicious test-time inputs.
- **Model reconstruction defenses** aim to cleanse the model on held-out clean validation data to remove any backdoor effect on the models.

For Backdoor Detection and Backdoor Mitigation defenses, we sample 50% of the test set as the defense set (12.5% of the train set size). We followed the original settings, but tuned certain hyperparameters for the defenses to adapt better to our dataset in terms of ACC and ASR.

## G.1  CLPBA under data augmentations.

As shown in Table 9, GM attack is robust to MixUp and CutMix. While Noising and Denoising partially mitigate the attack, the attacker can craft an adaptive attack that applies the same augmentation to the poison crafting process, improving the robustness of poison samples to augmentations.

**Evaluation metrics.** We adopt different sets of metrics to comprehensively evaluate CLPBA with filtering and firewall defenses:

For Filtering Defenses:

- **Elimination Rate (ER):** The percentage of poisoned samples that are correctly filtered.

Table 9: Performance of GM attack under augmentations.

|  | MixUp | CutMix | Noising | Denoising |
|---|---|---|---|---|
| GM | 77.2 | 95.7 | 64.4 | 39.3 |
| GM (with augment) | N/A | N/A | 96.4 | 99.3 |

775    • **Sacrifice Rate (SR):** The percentage of clean samples that are incorrectly filtered.

776 For Firewall Defenses:

777    • **True Positive Rate (TPR):** The percentage of trigger source-class samples that are correctly
778      filtered.

779    • **False Positive Rate (FPR):** The percentage of non-trigger samples that are incorrectly
780      filtered.

781 ## G.2    Filtering-based & Firewall defenses.

Table 10: Performance of GM attack under filtering defenses

| Filtering Defenses | | | | | |
|---|---|---|---|---|---|
| Metrics | AC | SS | DeepKNN | SPECTRE | CT |
| ER (%) | 0.0 | 61.7 | 23.3 | 91.7 | 48.3 |
| SR (%) | 7.6 | 32.6 | 0.0 | 30.4 | 4.94 |
| ASR (%) | 97.7 | 0.0 | 5.3 | 0.0 | 69.3 |

782 We evaluate CLPBA against 5 representative filtering-based defenses and six representative firewall
783 defenses:

784    • **Activation Clustering (AC)** [4]: This defense filters poisoned inputs in the latent space of the
785      poisoned model via clustering. It assumes that the poisoned inputs form a small cluster separate
786      from the clean inputs.

787    • **Spectral Signatures (SS)** [38]: This defense identifies a common property, spectral signature, of
788      backdoor attacks: Feature representations of the poisoned samples strongly correlate with the top
789      singular vector of the feature covariance matrix. This defense then filters a predefined number of
790      samples that have the highest correlation to the singular vector.

791    • **SPECTRE** [17]: This defense improves upon the Spectral Signature defense with robust covariance
792      estimation that amplifies the spectral signature of corrupted data.

793    • **DeepKNN** [31]: This defense was originally introduced for clean-label data poisoning. It assumes
794      that poisoned samples exhibit different feature distributions from clean examples in the feature
795      space. It then uses K-nearest neighbors to filter samples with the most number of conflicting
796      neighbors (neighbors that have different labels).

797    • **Confusion Training (CT)** [32]: This is a proactive defense technique that deliberately applies an
798      additional poisoning attack on an already poisoned dataset to actively disrupts benign correlations
799      in the data while amplifying the backdoor patterns, making them easier to detect.

800    • **STRIP** [13]: This defense detects poisoned inputs by applying random perturbations and observing
801      the model's prediction entropy. Poisoned inputs typically produce more consistent (lower entropy)
802      predictions under perturbations compared to clean inputs.

803    • **IBD-PSC** [19]: This defense clusters inputs based on their feature representations and identifies
804      poisoned samples as distinct clusters in the feature space, separate from clean samples.

805    • **Frequency-based Detection** [49]: This defense identifies backdoor triggers by analyzing frequency
806      patterns in the input data. Trigger artifacts often show statistically distinct patterns that can be
807      isolated through frequency domain analysis.

808    • **Cognitive Distillation (CD)** [20]: This method detects backdoor patterns by isolating minimal
809      features, called Cognitive Patterns (CPs), that trigger the same model output. Backdoor samples
810      consistently yield unusually small CPs, making them easy to identify.

- **SCALE-UP** [16]: This method etects backdoor inputs by checking for unusually consistent model predictions when input pixels are scaled. It works in a black-box setting without needing model access.

- **BadEXpert** [45]: This method creates a specialized "backdoor expert" model from the victim model to identify and filter poisoned inputs accurately, maintaining good clean-data performance.

**Evaluation on Filtering Defenses** As shown in Table 10, we find that most filtering-based defenses are not robust against our clean-label poisoning attack.

- **AC and DeepKNN** are largely ineffective, with Elimination Rates (ER) of 0.0% and 23.3%, respectively. This is because CLPBA is designed to make poisoned samples indistinguishable from benign samples in the feature space. The poisoned inputs are crafted to lie within the distribution of the target class, thereby violating the core assumption of these defenses that poisoned data will form separable clusters or have conflicting neighbors. Consequently, the Attack Success Rate (ASR) remains high at 97.7% against AC.

- **SS and SPECTRE**, which rely on spectral signatures, can successfully mitigate the attack, reducing the ASR to 0.0%. SPECTRE, in particular, identifies and removes 91.7% of the poisoned samples. However, this effectiveness comes at an unacceptably high cost: both defenses incorrectly filter over 30% of the clean samples (Sacrifice Rate, SR), rendering them impractical for real-world use. This indicates that while CLPBA leaves a detectable spectral artifact, it is not distinct enough to be separated from benign data without significant collateral damage.

- **Confusion Training (CT)** shows limited effectiveness. While it manages to filter nearly half of the poisoned samples (ER of 48.3%), the ASR remains high at 69.3%. This suggests that the backdoor patterns embedded by CLPBA are robust and not easily amplified or isolated by the disruptive signals introduced by CT.

In summary, CLPBA successfully evades defenses that assume feature-space separability and forces other methods like SPECTRE to discard an impractical amount of clean data to be effective.

**Evaluation on firewall defenses.** As demonstrated in Table 11, input-level detection methods are not effective for CLPBAs because they either miss trigger samples or incorrectly filter out too many benign samples. This behavior is expected, as CLPBAs challenge the main assumptions underlying these defenses:

Table 11: Performance of GM attack under Firewall defense.

| Metrics | Firewall Defenses | | | | | |
|---|---|---|---|---|---|---|
| | **STRIP** | **CD** | **IBD-PSC** | **Frequency** | **BadEXpert** | **SCALE-UP** |
| TPR (%) | 0.0 | 79.6 | 0.0 | 0.0 | 0.0 | 3.7 |
| FPR (%) | 6.7 | 65.3 | 16.7 | 0.9 | 16.7 | 25.1 |
| ASR (%) | 100.0 | 21.3 | 100.0 | 100.0 | 100.0 | 96.3 |

- **STRIP and IBD-PSC**: These methods assume that backdoor correlation (trigger and target label prediction) is more consistent than the classification of benign samples, and thus find ways to unlearn normal classification tasks to highlight trigger samples. However, since CLPBAs work by synthesizing natural features from the target class with the trigger, unlearning normal classification tasks also unlearns the backdoor correlation between the trigger and the target class.

- **Frequency-based Detection**: This method assumes that poisoned samples exhibit high-frequency artifacts that differ from benign ones. This assumption holds true for digital triggers, where there is no inherent correlation between the trigger and the natural image content in the pixel space. However, physical triggers, which are integrated naturally into the image, do not produce such high-frequency artifacts, making this defense less effective.

- **Cognitive Distillation**: This approach assumes that a backdoored model focuses on much smaller regions for classifying trigger samples than for classifying clean samples. However, because CLPBAs aim to embed the distribution of the source class with the trigger into the feature space of the target class, the classification region for trigger samples is larger. The model relies on a combination of the trigger and natural features of the source class for misclassification.

It is also important to note that these defenses are built for dirty-label all-to-one attacks instead of clean-label one-to-one attacks as CLPBA. Therefore, the assumption of a consistent backdoor correlation for STRIP and IBD-PSC may not hold true.

### G.3 Backdoor detection

**NC** [40], **ABS** [28]: NC uses an Anomaly Index metric to quantify how unusually small the reverse-engineered trigger perturbation for a given class is compared to others. Classes with high Anomaly Index values (greater than 2.0) are flagged as likely backdoor targets. However, in our experiments, NC only successfully identified the correct target class in 2 out of 10 trials. We attribute this limitation to NC's assumption of small, memorized trigger perturbations, which fails for CLPBA since it leverages adversarial feature manipulations rather than simple memorized trigger features (Section 5). ABS (Artificial Brain Stimulation) first identifies subsets of suspicious neurons and associates them with their suspected target classes by analyzing neuron activations. It then uses these identified neurons to reverse-engineer the backdoor trigger pattern. Despite this sophisticated approach, ABS fails against CLPBA because the malicious neurons it detects are consistently linked to incorrect target classes. Consequently, the synthetic triggers reverse-engineered by ABS yield a 0% attack success rate (ASR), in contrast to a 97.7% ASR achieved by the physical trigger.

### G.4 Backdoor mitigation

**NAD** [25], **I-BAU** [44]: NAD mitigates backdoors by fine-tuning the poisoned model on a clean defense dataset to construct a teacher model and then performing distillation onto the original poisoned model by matching activations in convolutional layers. I-BAU employs adversarial unlearning to remove backdoors by iteratively optimizing an implicit hypergradient objective. Our experiments demonstrate that NAD effectively defends against CLPBA without reducing clean accuracy (ACC), reducing ASR from 97.7% to 3.3%. In contrast, I-BAU is less effective against CLPBA, only decreasing ASR to 93.3% after 100 fine-tuning rounds.

