# OpenReview forum: "Clean-Label Physical Backdoor Attacks with Data Distillation"
_NeurIPS.cc/2025/Workshop/Reliable_ML — NeurIPS 2025 - Reliable ML Workshop_

### Official Review · Reviewer_5QD7 · 2025-09-14
**Review of NeurIPS 2025 Workshop Reliable ML Submission7 -  Clean-Label Physical Backdoor Attacks with Data Distillation**

**Rating:** 7
**Confidence:** 2

**Review:**

# Summary
This paper introduces the Clean-Label Physical Backdoor Attack (CLPBA), a novel and stealthy method for poisoning deep learning models.
In contrast to previous backdoor attacks based on digital triggers (special patterns digitally added at inference time to cause misclassification), an emerging line of work studies physical triggers (natural objects added to the physical environment, e.g., sunglasses, tennis balls that can be added naturally into a scene).
Unlike previous physical backdoor attacks that require mislabeling training images containing a trigger (a "dirty-label" approach), CLPBA works without manipulating labels or directly injecting the trigger into the training data, making it much harder to detect by human inspection.

## Contributions and Methods
The primary contribution is framing the backdoor attack as a "Dataset Distillation" problem. The core idea is to craft imperceptible perturbations and add them to a small number of correctly labeled target-class images. These perturbations are optimized to distill the essential features of a physical trigger (e.g., a specific pair of sunglasses) into the poisoned samples. When a model is trained on this data, it learns an association between the trigger and the target class without ever seeing the trigger in the training set.

The paper proposes three variants of this attack, each optimizing in a different space:
1. Parameter Matching (PM): Aims to match the training trajectory of a model trained on poisoned data with one trained on a traditional dirty-label dataset.
1. Gradient Matching (GM): Aligns the gradients produced by the poisoned samples with those from the trigger data.
1. Feature Matching (FM): Minimizes the difference between the feature representations of the poisoned images and the trigger images.

## Experiments
The authors created two new datasets to evaluate their method: a facial recognition dataset and an animal classification dataset, featuring various real-world physical triggers like sunglasses.

The key finding is that CLPBA can (surprisingly) achieve comparable or even better performance than dirty-label physical attacks.
Moreover, it seems to be decently robust against defenses.

# Strengths
- the proposed attack is novel and stealthier against human perception
- there is a nice theoretical motivation as a dataset distillation task
- experiments are reasonably extensive

# Weaknesses
- While it is probably standard in literature, the proposed attack requires access to the entire training dataset, which may not be practical.
- The paper notes that the Parameter Matching (PM) variant is less effective in some settings. The provided explanation is plausible but not empirically supported. A deeper analysis of why and when certain CLPBA variants fail would provide valuable insight.

# Suggestions
- It would be nice to mention if the ideas from CLPBA generalize to other data modalities.
- A small section of running times of the algorithm across datasets would be useful to gauge the practicality/scalability.
- Given the power of this attack, a dedicated section discussing the ethical implications and potential for misuse is warranted. Emphasizing that the goal of this research is to expose vulnerabilities to build more robust defenses would be a responsible addition.

---

### Official Review · Reviewer_cXAh · 2025-09-15
**Empirical results on backdoor attacks**

**Rating:** 7
**Confidence:** 3

**Review:**

# Summary

The paper extends the line of research on backdoor attacks, where the goal is to fool the model to misclassify the samples (from a source class to a target class) when a certain trigger is present. The paper introduces Clean-Label Physical Backdoor Attacks (CLPBA), which improves the previous work in several aspects:
CLPBA only needs to poison samples in the training set, instead of adding patterns at inference time.
The triggers will be physical, namely being natural objects in the environment (e.g., sunglasses).
The poisoned samples retain their original labels (clean-label), and do not contain a trigger explicitly but are perturbed with some noise.

The paper formalizes the attack goal as a Dataset Distillation problem. Given a dirty-label attacker, CLPBA will poison a subset of training samples by adding noise such that the model trained on the poisoned samples is close to the model trained on the dirty-label samples. The paper proposed three variants of CLPBA with different closeness measures, Parameter Matching (PM), Gradient Matching (GM) and Feature Matching (FM).

As a result, CLPBA closely matches and even surpasses in some settings the performance of dirty-label attack. Also, CLPBA shows robustness against representative defenses to backdoor attacks.


# Strengths

The authors make use of the dirty-label attacker as a reference to add perturbation so that the objective is only to match the training process on the perturbed samples with that on the dirty-label samples. This is a smart surrogate of the performance of the attacker, as one would expect the best a clean-label attacker can do is to match the performance of a dirty-label attacker (neglecting the possible memorization issue with dirty-label).


# Weaknesses / Limitations

The authors mention that feature matching estimates the difference between the poisoned samples and the source trigger distribution in a low-dimensional embedding space via a feature extractor $f$, possibly a neural network. It is not clear to me how they choose such a neural network in the experiment and how is the trade-off between efficiency and accuracy when the choice varies, as FM has the best performance under some settings.

Also, the authors choose a specific poison ratio $\alpha = 10\%$. It would be good if comparison between different ratios can be added in the experiment to verify the intuition that the performance improves with larger poison budgets.

# Suggestions

Several typos:
In line 133, $\bigcup_{i=1}^C$ should be $\bigcup_{c=1}^C$.

In line 157, the first loss should be $\mathcal{L}^{P_t(\delta)}(\theta)$.

Line 207 is the first time ASR is mentioned, but the explanation of the abbreviation is in line 247, which is confusing for the first time read. Also, it would be good if the definition of ASR can be added as it is an important metric.

In line 299, “Figure 2” should be “Table 2”.